# Cardiac Magnetic Resonance Imaging (CMRI) Applications in Patients with Chest Pain in the Emergency Department: A Narrative Review

**DOI:** 10.3390/diagnostics13162667

**Published:** 2023-08-14

**Authors:** Hossein Zareiamand, Amin Darroudi, Iraj Mohammadi, Seyed Vahid Moravvej, Saba Danaei, Roohallah Alizadehsani

**Affiliations:** 1Department of Cardiology, Faculty of Medicine, Sari Branch, Islamic Azad University, Sari 48161-19318, Iran; h_v.h_zarei@iausari.ac.ir; 2Student Research Committee, Sari Branch, Islamic Azad University, Sari 48161-19318, Iran; a.darroudi@hums.ac.ir; 3Department of Basic Sciences, Faculty of Medicine, Sari Branch, Islamic Azad University, Sari 48161-19318, Iran; imohammadi@farabi.tums.ac.ir; 4Department of Computer Engineering, Isfahan University of Technology, Isfahan 84156-83111, Iran; sa.moravvej@alumni.iut.ac.ir; 5Adiban Institute of Higher Education, Garmsar 35881-43112, Iran; ssabadanaei@gmail.com; 6Institute for Intelligent Systems Research and Innovation (IISRI), Deakin University, Waurn Ponds, Geelong, VIC 3216, Australia

**Keywords:** cardiac magnetic resonance imaging, emergency department, chest pain, heart disease

## Abstract

CMRI is the exclusive imaging technique capable of identifying myocardial edema, endomyocardial fibrosis, pericarditis accompanied by pericardial effusions, and apical thrombi within either the left or right ventricle. In this work, we examine the research literature on the use of CMRI in the diagnosis of chest discomfort, employing randomized controlled trials (RCTs) to evaluate its effectiveness. The research outlines the disorders of the chest and the machine learning approaches for detecting them. In conclusion, the study ends with an examination of a fundamental illustration of CMRI analysis. To find a comprehensive review, the Scopus scientific resource is analyzed. The issue, based on the findings, is to distinguish ischemia from non-ischemic cardiac causes of chest pain in individuals presenting with sudden chest pain or discomfort upon arrival at the emergency department (ED). Due to the failure of conventional methods in accurately diagnosing acute cardiac ischemia, individuals are still being inappropriately discharged from the ED, resulting in a heightened death rate.

## 1. Introduction

Chest pain is a commonly encountered ailment in the ED. The prompt and precise differentiation between grade 1 acute myocardial infarction (AMI) and other causes of myocardial injury is crucial for individuals with abnormal troponin levels [1]. The notable capability of high-sensitivity troponin tests to predict the absence of AMI highlights the significance of diagnostic methods in identifying the root cause of AMI [2]. CMRI is a valuable technique for stratifying patients in the ED who experience chest discomfort. Unlike invasive coronary angiography, CMRI is a non-invasive procedure that is less expensive and results in a shorter hospitalization duration. Additionally, CMRI provides significant information about the structure, function, tissue edema, and the location and nature of tissue damage in the heart, all of which can assist in determining various etiologies of cardiac injury. CMRI may assist in discriminating between chest symptoms caused by type 1 AMI and supply–demand imbalances caused by acute cardiac non-coronary artery disease. After conducting a comprehensive research review, it was determined that utilizing CMRI with stress monitoring is a safe approach for patients who arrive at the ED with chest pain, regardless of the varying troponin levels [3]. Combined positron emission tomography (PET) and CMRI integrates the advances in functional and high-quality anatomical imaging from these systems for many clinical and scientific studies, including a wide use in cardiology. In this combined scanner, motion fields of the heart organ are estimated from concurrently acquired tagged magnetic resonance images [4]. CMRI is a helpful, secure, cost-efficient, and effective alternative to conventional diagnostic techniques utilized in this patient group. It is a helpful technique for risk stratifying individuals with suspected heart pathology and confirming diagnoses without the need for invasive testing [5].

CMRI can be used in a wide range of clinical situations. Cardiac cine imaging may be employed to measure global and regional operations using a variety of approaches. An example of this is T2-weighted imaging, which can evaluate stress-induced myocardial edema, rest perfusion, and the presence of myocardial ischemia following the administration of a vasodilator-like adenosine, dipyridamole, or regadenoson [6]. The utilization of delayed-enhancement CMRI allows for the identification and characterization of myocardial necrosis, inflammation, and infiltrative diseases by assessing their presence and pattern [7,8,9,10]. Techniques for tissue description can be utilized as appropriate, including tomographic images for chest morphology, velocity flow projection for valve appraisal, and tissue characterization techniques (such as T1 parametric projection for infiltrative disease and T2* imaging for iron overburden) [11]. CMRI plays a crucial role as a diagnostic tool in patients diagnosed with myocardial infarction with non-obstructive coronary arteries (MINOCA). CMRI is a critical diagnostic tool for assessment of the existence of regional wall motion anomalies, tissue edema, and subendocardial late gadolinium enhancement (LGE) in a typical coronary distribution area [12,13,14]. In a patient with acute chest discomfort and high-sensitivity troponin, the diagnosis of AMI could serve as a powerful tool in chest pain patients in the ED and improve the effectiveness of patient management. MINOCA can manifest in 5% to 6% of cases, even when all the universally recognized criteria for a type 1 AMI are satisfied [15]. Thrombosis, superimposed thromboembolism, vasospasm, plaque disruption, takotsubo cardiomyopathy, myocarditis or a composite of such occurrences are all possible causes of MINOCA [16].

Nevertheless, since the coronary arteries may seem ordinary or almost standard around the period of angiography, the ischemia system of myocardial necrosis cannot be correctly diagnosed, and consequently, essential secondary preventive treatments (as an example, statins and antiplatelet drugs) may not be implemented. Whilst CMRI cannot distinguish between different causes of ischemic damage in patients with MINOCA (such as spasm or emboli), it is capable of detecting ischemia and distinguishing it from myocarditis, takotsubo cardiomyopathy, or other causes of acute cardiac noncoronary artery diseases (CNCDs) [17]. Relying solely on the absence of epicardial stenosis of the internal carotid artery is insufficient to determine the underlying cause of damage in a patient with abnormal troponin levels. Additional information from CMRI is necessary to make a conclusive assessment. To differentiate between actual ischemic injury (referred to as True-MINOCA) and myocardial injury unrelated to ischemia caused by acute CNCDs (referred to as False-MINOCA), CMRI is employed [5].

While computed tomography (CT) is commonly favored in emergency departments due to its speed and versatility, the unique advantages of CMRI should not be overlooked. CMRI provides non-invasive imaging, cost-effectiveness, and valuable information about the cardiac structure and tissue damage, aiding in accurate diagnosis and patient management. By incorporating machine learning and combining CMRI with stress monitoring or other imaging techniques, like PET, clinicians can enhance risk stratification and confirm diagnoses without invasive procedures. Considering these benefits, CMRI plays a crucial role in evaluating cardiac conditions in emergency departments [5].

CMRI offers numerous advantages, such as non-invasiveness, lower cost, and valuable information on a heart’s structure and tissue damage, making it a valuable tool in determining the etiology of cardiac injury. CMRI, when combined with stress monitoring or other techniques like PET, allows for risk stratification and confirmation of diagnoses without the need for invasive testing. It can assess myocardial edema, perfusion, necrosis, inflammation, and other characteristics, playing a crucial role in diagnosing conditions like MINOCA. Therefore, CMRI is an essential adjunct to accurately diagnose and manage patients, particularly in cases where the troponin levels are abnormal but epicardial stenosis is absent. CMRI offers numerous advantages, making it a powerful and versatile imaging technique in the diagnosis and management of cardiac injuries and related conditions. One of its primary strengths lies in its non-invasiveness, which minimizes patient discomfort and reduces the risk of complications associated with invasive procedures. Additionally, compared to other imaging modalities, CMRI tends to be more cost-effective, making it accessible to a broader range of patients. The detailed information obtained through CMRI about the heart’s structure and tissue damage is invaluable for physicians in determining the underlying cause of cardiac injury. It allows for a comprehensive assessment of various cardiac parameters, including myocardial edema, perfusion, necrosis, inflammation, and other key characteristics. This wealth of information aids in the accurate diagnosis of complex conditions like MINOCA, where traditional diagnostic methods may fall short. An essential aspect of CMRI’s utility is its versatility in combination with other diagnostic tools. When integrated with stress monitoring or techniques like PET, CMRI enables risk stratification and confirmation of diagnoses without resorting to invasive testing. This capability proves particularly valuable in cases where clinicians need to assess the extent of cardiac damage and the potential risks posed to the patient’s health [17].

## 2. Application of CMRI in the Diagnosis of Different Types of Chest Pains

CMRI provides a wide range of different diagnoses unrelated to acute coronary syndrome (ACS) that can effectively explain the symptoms, and it also includes incidental findings. The identification of these unforeseen findings could have implications for patient management, leading to the establishment of new diagnoses or the need for further investigations. Between 2011 and 2015, adult patients with suspected ACS who visited an academic ED showed no indications of ischemia on initial electrocardiogram (ECG), had a minimum of one negative cardiac biomarker, and then had CMRI as a component of their diagnostic assessment were prospectively recruited. This finding suggests that CMRI can be used to diagnose symptomatic coronary artery disease (CAD) and potentially non-CAD severe cardiac abnormalities. These considerations may influence its usage in ACS workups in the emergency department [18]. In the investigation of the prognostic and diagnostic utility of CMRI in the diagnosis of ischemic heart disease (IHD), researchers have explored the current improvements, limitations, and future directions. For example, Fagiry et al. examined these aspects to enhance the effectiveness of CMRI in clinical practice. For this upcoming study, a group of 100 individuals clinically diagnosed with ischemic heart disease (IHD) were selected as participants. The findings of this study showed that while CMRI is a comprehensive prognostic and diagnostic tool for assessment of LV function, myocardial perfusion, viability, and coronary anatomy, in the diagnosis of IHD in the patients, it has a sensitivity, specificity, and accuracy of 97%, 33.33%, and 95.15%, respectively [19].

An inherent problem associated with cardiac catheterization and CT coronary angiography is the considerable radiation exposure endured by the patient during the procedure. Consequently, employing CMRI technology to solve such problems is highly useful to patients [20]. Jalnapurkar et al. [21] conducted a study to examine the diagnostic importance of stress CMRI in women presenting with suspected ischemia. The study focused on 113 female patients who underwent stress CMRI, encompassing anatomic, functional, adenosine stress perfusion, and delayed-enhancement photography. Prior to this, these patients had undergone assessment for indications and manifestations of ischemia; however, there was no indication of obstructive CAD detected. From 113 patients, 65 were diagnosed with coronary microvascular dysfunction (CMD) on the basis of subendocardial perfusion abnormalities consistent with myocardial ischemia on stress CMRI, 10 with CAD, 2 with left ventricular (LV) hypertrophy, and 3 patients were diagnosed with congenital coronary anomalies or cardiomyopathy that had not been detected in prior cardiac evaluations. The rest (33 patients) were normal. These findings indicate that stress CMRI often reveals abnormalities and offers diagnostic value in identifying CMD in women who display symptoms and indications of ischemia but do not show any signs of obstructive CAD. Stress CMRI appears helpful for diagnostic assessment in these diagnostically challenging people.

To detect LGE patterns via cardiac MRI in high-risk patients with right ventricular dysfunction following the placement of a left ventricular assist device (LVAD), Simkowski et al. [22] proposed an unsupervised machine learning (ML) method. They utilized the 17-segment model to extract LGE patterns from CMRI scans performed on patients who had received an LVAD at a medical facility within a 12-month timeframe. Employing an unsupervised ML technique for hierarchical agglomerative clustering, the patients were subsequently classified based on similarities in the LGE patterns. The clusters which resulted from this were then statistically compared. Based on the findings, the application of unsupervised ML to analyze the LGE patterns observed on CMRI has the capability to identify groups of patients who are prone to developing right ventricular failure (RVF). Patients diagnosed with non-ischemic and mixed etiologies of heart failure may face an increased risk of developing RVF compared to those with purely ischemic causes. This heightened risk can be attributed to the extensive involvement of biventricular myocardium indicated by the observed LGE patterns on CMRI. Alsunbuli [23] evaluated several imaging modalities by using their inherent features to benchmark against a simulated ideal test, utilizing a qualitative approach to the comparison, as well as the various societies’ guidelines. According to the findings, CMRI poses no danger of radiation exposure but provides lesser resolution than CT. It requires more time from physicians and patients and hence is more demanding. It requires fewer operators than echocardiography and enables the identification of small changes in serial follow-up evaluations, particularly for the LV volume and function. CMRI can also be used during pregnancy. In terms of the drawbacks, it cannot be used intra-procedurally and is contraindicated in the presence of certain pacemakers. CMRI is also susceptible to artifacts that may be detected using a chest X-ray, such as a retrocardiac surgical needle in one instance.

Grober et al. [24] conducted a comparison between diffusion-weighted MRI (DMRI) and conventional MRI techniques to detect microadenomas in patients with Cushing’s disease. They further evaluated the efficacy of a 3D volumetric interpolated breath-hold examination, a 3D T1 sequence known as a spoiled gradient echo (SGE), which offers enhanced soft-tissue contrast and improved resolution. SGE has better sensitivity for identifying and localizing pituitary microadenomas than DMRI. However, DMRI is rarely used to diagnose adenoma. SGE should be included in the routine MRI procedure for Cushing’s disease patients. Moonen et al. [25] used CMRI to determine the frequency of Fabry disease in a group of individuals with inexplicable LGE. Fabry disease is a rare X-linked genetic condition with cardiac symptoms such as LVH, contractile failure, and fibrosis, which can be seen as LGE of the myocardium on CMRI. Fabry disease is a critical diagnosis to establish, since the missing enzyme can be replaced for the rest of one’s life. In terms of detecting and pinpointing pituitary microadenomas, the SGE technique demonstrates greater sensitivity compared to DMRI. It is uncommon for an adenoma to be solely detected using DMRI. Therefore, it is recommended to include SGE as a standard component of the MRI protocol for patients diagnosed with Cushing’s disease. In a study conducted by Moonen et al. [25], the presence of Fabry disease was examined in a cohort of patients displaying unexplained LGE on CMRI. Diagnosing Fabry disease holds significant importance due to the availability of lifelong enzyme replacement therapy as a treatment option for the deficient enzyme. According to the findings, the presence of unexplained LGE on CMRI could potentially indicate the presence of late-onset Fabry disease.

Figure 1 depicts the chest pain approach utilized in the ED. A nurse performs triage targeted toward the main complaint throughout the triage screening procedure. The person’s signs and symptoms, onset, personal history, drugs taken, and allergies are all discussed. The existence of breathing and pulse, as well as the detection of circumstances that indicate a high risk of mortality, are all evaluated. When a patient complains of chest discomfort, they are referred for an ECG. Following that, the medical team evaluates the patient and prescribes the appropriate treatment. The risk stratification is divided into five stages. This technique was characterized as positive based on the American Heart Association criteria when the patient was assessed as a high priority.

The studies conducted on the utilization of CMRI in identifying chest discomfort caused by various disorders have been gathered in Table 1.

## 3. Utilizing Machine Learning for the Diagnosis of Chest Pain through CMRI

Artificial intelligence (AI) and machine learning (ML) are quickly gaining traction in medicine [19,20]. In the coming years, they are anticipated to profoundly change clinical practice, notably in the field of medical imaging [4,29]. AI is a broad term that refers to using robots to do activities that are common to human intellect, such as inferring conclusions through deduction or induction. On the other hand, ML is a more limited kind of computer processing that learns how to generate predictions using a mathematical model and training data. By being subjected to more instances, ML learns parameters from examples and can perform better at tasks like identifying and distinguishing patterns in data. The most sophisticated ML techniques, also known as DL, are particularly well-suited for this task. DL segmentation methods have recently been proven to outperform classic methods such as cardiac atlases, level set, statistical models, deformable models, and graph cuts. Nevertheless, a recent study of a number of automated techniques revealed that in more than 80% of CMRIs, even the top performing algorithms produced anatomically implausible segmentations [30]. When specialists perform segmentation, such mistakes do not occur. To gain acceptability in clinical practice, the automated methods’ flaws must be addressed through continued research. This can be accomplished by producing more accurate segmentation results or developing techniques that automatically detect segmentation errors.

By combining automated segmentation and evaluating segmentation uncertainty, Sander et al. employed CMRI to identify regions in the images where local segmentation failures occur. They utilized a convolutional neural networks (CNNs) uncertainty to discover local segmentation problems that may require expert repair. To compare the performance of manual and (corrected) automatic segmentation, the Dice coefficient, 3D Hausdorff distance, and clinical markers were utilized. The findings suggest that combining automated segmentation with manual correction of identified segmentation errors results in enhanced segmentation accuracy and a significant 10-fold decrease in the time required by experts for segmentation compared to manual segmentation alone, as demonstrated in the studies [31]. During segmentation training, Oktay et al. [32] devised an auto-encoder-based anatomically restricted neural network (NN) that utilizes constraints to make inferences about limitations. In a study, Duan et al. [33] incorporated atlas propagation to explicitly enforce shape refinement in a DL-based segmentation approach for CMRIs. This was extremely convenient when there were photo capture artifacts present. By employing cardiac anatomical metrics, Painchaud et al. [34] devised a post-processing technique to identify and transform anatomically questionable heart segmentations into accurate ones. Employing an ML-based method, Park et al. [35] predicted AMI. The occlusion of coronary arteries is responsible for the occurrence of AMI, and prompt revascularization is necessary to improve the prognosis. However, AMI has been misdiagnosed as other illnesses, and reperfusion delay has been linked to a poor outcome in patients. The authors used ML algorithms to anticipate AMI in patients with acute chest discomfort based on data collected at admittance. The best area under the curve was obtained in this research, demonstrating that ML is a more powerful technique for AMI prediction. Although the fast growth of ML offers many benefits, there are still several issues in relation to large-scale clinical use [36]. CMRI includes a lot of scanning layers and a complicated process, thus there are certain to be some low-quality pictures.

Controlling the quality of cardiovascular images is so critical. Varied manufacturers, various machine types, and different MCE scanning parameters all influence ML. At the moment, the ML system based on DL suffers from a lack of explainability. After much training, an ML model could identify myocardial fibrosis based on a picture. However, it might not explain what effective characteristics it learned to reach such a conclusion. As a result, explainability is a vital study topic concerning medical ML [37,38]. To ensure high accuracy and optimize the algorithm, it is crucial to utilize a substantial amount of high-quality data during the initial learning phase of ML, leveraging its inherent capabilities. Furthermore, the acquisition cost and time required to cardiovascular imaging, particularly MCE data from CMRI, are significant. The critical task at hand is to establish a model that can effectively learn the best optimal solution even when provided with limited samples. Through the utilization of migration learning, it becomes feasible to transfer valuable information from prior ML models to novel models, resulting in a reduction in the required data resources for DL [39].

Table 2 outlines numerous CMRI machine-learning applications for diagnosing chest discomfort based on various experiments accomplished by various authors in various years.

## 4. Research Statistics

This section presents the statistics and numbers of papers published in the Scopus journals between 2012 and 2021. To analyze the number of papers and find the research trend in this field, an algorithm for data summarization is utilized.

To start the evaluation of studies, the related keywords are chosen. The initial keywords are “chest pain” and “CMRI,” in which you can search the keywords with “+” symbols. Then, limit the research based on years of publication. After analyzing the studies, new keywords are identified based on word frequency. Some additional keywords include “heart disease”, “machine learning”, and “image processing”. These keywords are also analyzed in combination with the “CMRI” keyword.

The number of articles in this range is from 132 to 185, which has shown a steady increase. The highest amount belongs to 2020, with 185 articles, and the lowest amount belongs to 2016, with 132 articles. This indicates growing interest in the field and highlights the importance of research in understanding and diagnosing chest pain using CMRI techniques.

The research papers constitute the highest proportion among the different types of studies. This suggests that a significant amount of research in this field is dedicated to presenting original findings and contributing to the existing body of knowledge. Additionally, other types of publications, such as reviews, chapters, and conference papers, also contribute to the dissemination of information in this domain.

Furthermore, the research is categorized based on the fields of study, providing insights into the interdisciplinary nature of CMRI research. The majority of researchers, approximately 74%, belong to the medical field. This reflects the significance of CMRI in the medical community for diagnosing and understanding chest pain and related conditions. Additionally, about 8% of the research is related to biochemistry, genetics, and molecular biology, indicating the importance of studying the underlying molecular mechanisms associated with chest pain.

Based on the analysis, several countries stand out as leading contributors to the CMRI research. The United States, United Kingdom, Germany, and Canada have the highest number of publications, showcasing their active involvement and research output in this field.

## 5. Implementation Results

We offer a tiny example of implementation in this section of the study to provide a clearer overview of CMRI use in the ED. Late enhancement imaging tests were performed 15 min following gadolinium–DTPA injection utilizing a 3D-gradient faulty turbo fast-field echo (FFE) sequence that includes an individually designed 180° inversion pre-pulse (Look-Locker) to provide appropriate myocardium suppression [47]. A series of images were acquired using a 2D-sequence approach, which included short-axis images with a 5 mm slice thickness encompassing the entire left ventricle, along with two to three long-axis views. The presence of dark patches within the enlarged myocardium supplied by the infarct artery indicated the presence of persistent microvascular obstruction. Various patterns of late enhancement, including subendocardial, transmural, intramural, subepicardial, and diffuse patterns, were detected (Figure 2). To visualize cardiac edema, a T2-weighted turbo spin-echo sequence was employed, along with a fat saturation pulse. Images were acquired in a continuous short-axis orientation, covering the entire left ventricle, with a slice thickness of 15 mm. Myocardial edema was defined as a relative myocardial EPCs intensity exceeding 2.0 times that of skeletal muscle.

Coronary artery disease leads to myocardial damage, which can be identified through subendocardial or transmural late enhancement patterns. In contrast, acute myocarditis is often associated with the presence of late enhancement, as characterized by a diffuse, intramural, or subepicardial pattern. Patients with ST-segment elevation myocardial infarction (STEMI) had the greatest levels of creatine kinase (CK), troponin-I, and leukocytes. They gradually dropped from individuals with non–ST-segment elevation myocardial infarction (NSTEMI), acute myocarditis, and takotsubo cardiomyopathy to takotsubo cardiomyopathy (Figure 3). In terms of the C-reactive protein (CRP) levels, patients with acute myocarditis exhibited the highest initial and peak values. There were statistically significant differences in the levels of CK, troponin-I, and the first CRP among the different groups.

The volumes and ejection percentages of the ventricles were considerably different. Acute myocarditis patients had the greatest LV volumes. The LV ejection fraction of STEMI patients was considerably lower than that of NSTEMI patients (*p* = 0.006). Acute myocarditis patients had substantially greater RV volumes than other categories (*p* = 0.03). In the group of patients experiencing their first episode of severe chest pain, wall motion abnormalities were detected in all 95/95 (100%) cases of STEMI, 51/68 (75%) cases of NSTEMI, 18/27 (66.7%) cases of acute myocarditis, and 12/12 (100%) cases of takotsubo cardiomyopathy. The observed differences were statistically significant (*p* < 0.001). A random distribution of wall motion anomalies was seen in individuals with acute myocarditis. The aberrant wall motion in individuals with takotsubo cardiomyopathy was concentrated in the midventricular–apical regions.

## 6. Discussion

This research work focuses on investigating the literature concerning the utilization of CMRI for diagnosing chest discomfort. The study encompasses a comprehensive analysis of chest disorders and explores the application of machine learning techniques in their detection. Moreover, the research concludes by providing a detailed illustration of the fundamental aspects of CMRI analysis. To ensure a thorough investigation, the Scopus scientific resource was extensively reviewed, allowing for a comprehensive examination of the topic. The primary concern addressed in this study is the differentiation between ischemic and non-ischemic cardiac causes of chest pain in individuals who present with sudden chest pain or discomfort upon their arrival at the ED. Conventional diagnostic methods struggle with accurately diagnosing acute cardiac ischemia, causing inappropriate discharge and increased mortality rates. CMRI can enhance accuracy and prevent misdiagnoses, emphasizing the importance of effective utilization. The overall objective of the study is to improve the diagnosis of chest discomfort by boosting CMRI’s capacity to identify abnormalities and investigating machine learning techniques. By addressing the shortcomings of traditional diagnostic techniques, it seeks to improve patient outcomes, lower death rates, and enhance cardiology.

This article has the following limitations, despite its positive aspects:Selection bias: The article does not provide details about the criteria used to select the studies included in the research review. It is important to consider that studies with positive results may be more likely to be published, while studies with negative or inconclusive results may be overlooked. This selection bias can lead to an overestimation of the effectiveness of CMRI in diagnosing chest discomfort.Interpretation bias: CMRI interpretation requires expertise and subjective judgment. The article does not mention whether the researchers or reviewers were blinded to the clinical information. If they were not blinded, knowledge of the patient’s clinical status or symptoms could introduce bias into the interpretation of CMRI findings.Interobserver variability: Different observers’ interpretations of CMRI may differ. The study offers no inquiry into whether the analysis employed several reviewers or whether steps were taken to evaluate and reduce interobserver variability. The trustworthiness of the study’s findings may be impacted by different reviewers’ inconsistent interpretations of CMRI data.Lack of gold standard comparison: Although the article cites the use of CMRI as a substitute for traditional diagnostic methods, it supplies no information regarding the reference standard or gold standard that was used to determine the accuracy of CMRI. It is difficult to adequately assess the genuine diagnostic performance of CMRI without a direct comparison to a recognized gold standard.Generalizability: The study populations’ characteristics or the environments in which the investigations included in the research review were carried out are not described in the article, which limits its potential to generalize. The findings’ applicability to other patient demographics or healthcare environments must be taken into account. Depending on the patient’s demographics, comorbidities, and access to knowledge and resources, CMRI may or may not be useful for detecting chest discomfort.Potential conflicts of interest: Conflicts of interest that might have existed between the researchers or the funding sources are not addressed in the publication. Financial ties to businesses that make CMRI equipment or drugs related to it can skew the results of studies. Any conflicts of interest must be disclosed in order to maintain transparency and reduce potential bias.In addition to the previously mentioned limitations, it is important to address the practical applicability of cardiac MRI in daily clinical practice:Feasibility in emergency settings: The paper does not thoroughly discuss the feasibility of performing cardiac MRI in emergency departments or emergency rooms (ED/ER). Given the time-consuming nature of cardiac MRI, it may not be practical to perform this imaging modality in acute situations where timely interventions are crucial.Resource utilization: Cardiac MRI requires specialized equipment, trained personnel, and dedicated facilities. Assessing the availability and allocation of these resources, as well as their cost-effectiveness, is crucial in understanding the practicality and sustainability of widespread cardiac MRI implementation.Patient selection criteria: Not all patients with chest discomfort or suspected cardiac conditions may be suitable candidates for cardiac MRI due to factors such as contraindications, patient stability, or the urgency of intervention. Understanding the limitations of and specific indications for cardiac MRI in the emergency setting is essential for its optimal utilization and decision-making.
Limited evidence in the acute setting: The article predominantly focuses on studies conducted in the chronic/subacute setting, where patients were likely admitted to the ward. The lack of effective data on the clinical application of MRI in the acute setting raises concerns about the generalizability of the findings to emergency situations. The article should acknowledge the limitations of the available evidence and highlight the need for further research specifically targeting the acute setting. The generic messages derived from predominantly chronic/subacute studies may not be directly applicable or reproducible in acute clinical scenarios.

In order to advance the field of CMRI and its application in diagnosing chest discomfort, several key areas for future research have been identified. One area of focus is the refinement of machine learning approaches. By leveraging the power of artificial intelligence, researchers aim to develop more robust algorithms that can analyze CMRI data with increased accuracy and efficiency. This would enable clinicians to obtain more reliable and timely diagnostic information, leading to improved patient outcomes.

Another crucial aspect is validating the effectiveness of CMRI in real-world clinical settings. While CMRI has shown promise in research studies, it is essential to assess its performance in everyday clinical practice. By conducting large-scale studies and comparative analyses, researchers can gather valuable insights into CMRI’s diagnostic capabilities and identify any limitations or challenges that need to be addressed.

Exploring advanced imaging techniques is also a priority for future research. This includes investigating new CMRI sequences and protocols that can provide even more detailed and comprehensive cardiac information. By pushing the boundaries of imaging technology, researchers can potentially identify subtle cardiac abnormalities that may have been previously missed, thereby improving diagnostic accuracy.

Furthermore, improving the diagnosis and management of MINOCA is an important area of research. CMRI has shown promise in identifying the underlying causes of MINOCA, such as myocarditis or microvascular dysfunction. Future studies should aim to refine the CMRI protocols specifically tailored for MINOCA diagnosis, leading to more personalized and effective management strategies.

Integration of CMRI with other imaging modalities is also a promising avenue for future research. By combining the strengths of CMRI with other imaging techniques, such as PET or coronary angiography, a more comprehensive assessment of cardiac function and perfusion can be achieved. This multimodal approach has the potential to provide a more holistic understanding of cardiac conditions, aiding in treatment planning and monitoring long-term outcomes.

## 7. Conclusions

The management of patients with chest pain or discomfort is a common and challenging clinical problem. In this paper, the review of CMRI research highlights its practical implications for emergency department management, providing comprehensive information on cardiac structure, tissue damage, and myocardial fibrosis. Machine learning methods, particularly deep neural networks, have potential for accurate diagnosis and treatment planning. Future CMRI advancements aim to develop accurate, adaptable methods for routine clinical applications, ensuring efficiency in emergency settings. This involves the academic community, healthcare institutes, and medical imaging industry integrating research findings. CMRI and machine learning advancements improve patient care and decision-making in emergency departments. Such methods offer detailed assessment of cardiac conditions, improving risk stratification and accurate diagnoses. CMRI’s evolution will significantly impact emergency department management of chest pain patients.

To enhance the credibility of future research, it is essential to prioritize the inclusion of RCTs and prospective studies specifically conducted in acute settings, focusing on individuals presenting with sudden chest pain or discomfort upon arrival at the ED. RCTs play a crucial role in providing a more robust assessment of CMRI’s diagnostic accuracy and its potential utility in promptly diagnosing chest discomfort in acute cases. By conducting well-designed RCTs, researchers can effectively compare CMRI’s performance against other imaging techniques and conventional diagnostic methods, thus yielding more reliable and reproducible findings. These comprehensive RCTs have the potential to significantly aid in accurately differentiating between ischemic and non-ischemic cardiac causes of chest pain, ultimately leading to improved patient outcomes. Moreover, the integration of CMRI in the acute diagnostic pathway has the potential to reduce the rate of inappropriate discharges from the ED, which, in turn, can contribute to a lower mortality rate associated with undiagnosed cardiac conditions.

## Figures and Tables

**Figure 1 diagnostics-13-02667-f001:**
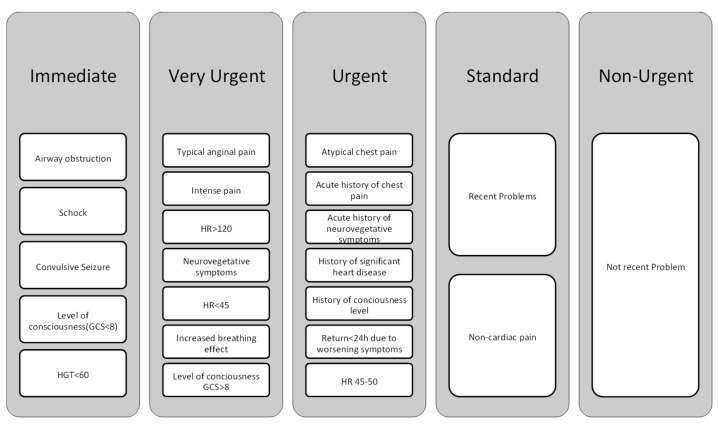
The chest pain protocol used in the emergency department.

**Figure 2 diagnostics-13-02667-f002:**
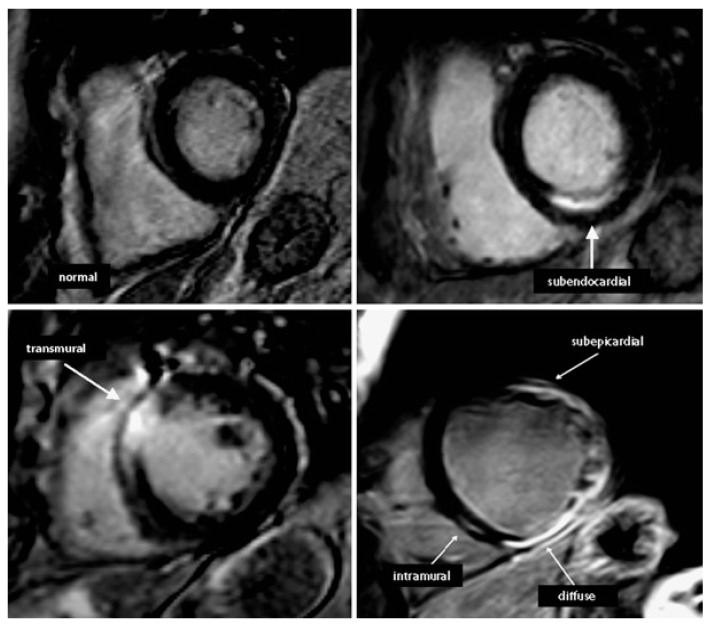
Illustrations of various late enhancement patterns.

**Figure 3 diagnostics-13-02667-f003:**
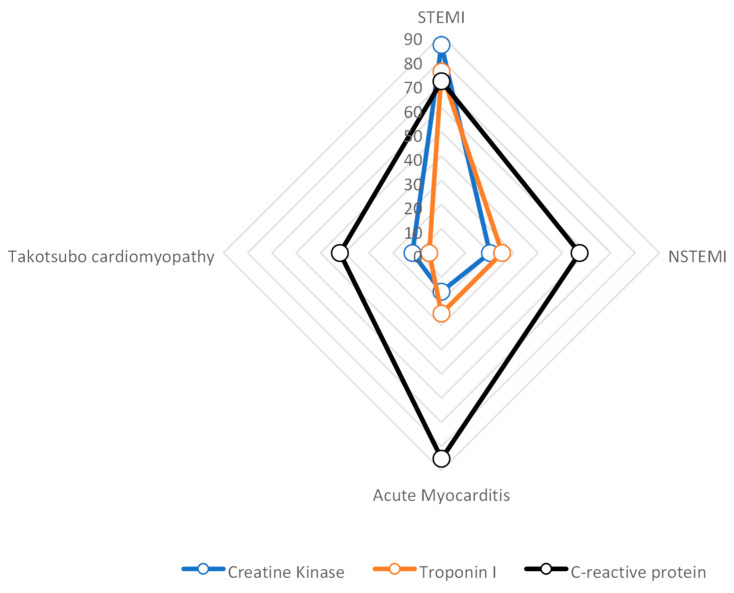
Mean maximal values of CK (based on 3000 U/L), troponin-I (based on 40 µg/L), and CRP (based on 70 mg/L) for patients with STEMI, NSTEMI, acute myocarditis or takotsubo cardiomyopathy.

**Table 1 diagnostics-13-02667-t001:** Application of CMRI in the diagnosis of chest pain related to disease.

Author/Year	Disease	Design of the Study	Number of Patients	Region	Aim	Results
Fagiry et al. [19], 2021	Ischemic heart disease	The study was conducted at the Radiology and Medical Imaging Department, King Fahad Medical City (KFMC), Riyadh, Saudi Arabia.	This study was designed as a prospective cohort study	The study enrolled 100 patients with ischemic heart disease (IHD)	Explore CMRI’s usefulness in IHD diagnosis	CMRI showed sensitivity, specificity, and accuracy in diagnosing IHD compared to angiography
Simkowski et al. [22], 2020	RVF	The study utilized a retrospective observational design	The study included a total of 31 patients who underwent pre-operative CMRI prior to receiving a LVAD	Not mentioned	Apply ML technique to detect LGE patterns in RV dysfunction post LVAD implantation	No significant association found between RVF and LGE in RV myocardium or insertion sites
Alsunbuli [23], 2020	Congenital heart disease	The study utilized a qualitative approach to compare different imaging modalities for grown-up congenital heart diseases (GUCH); it involved a systematic search for evidence from resources such as PUBMED, EMBASE, ScienceDirect, CINAHL, NICE, ESC, ACC/AHA; the studies identified were analyzed for inclusion in the body of evidence	Not mentioned (the study focused on comparing different imaging modalities rather than involving a specific patient cohort)	Not mentioned	Compare imaging modalities and their limitations	CMRI is valuable for diagnosis and follow-up but has limitations with certain conditions and cost
Moonen et al. [25], 2020	Anderson– Fabry disease	A cohort study including 16+ patients with left ventricular hypertrophy, idiopathic dysfunction, and arrhythmia; patients with unexplained late gadolinium enhancement were tested for Fabry disease using genetic testing or dried blood spot test	The study included a total of 79 patients with unexplained LGE on CMRI	Not mentioned	Investigate frequency of Anderson–Fabry disease in people with LGE on CMRI	Patchy mid-wall augmentation observed in inferoseptum region in people with LGE on CMRI
Groepenhoff et al. [26], 2020	Coronary vascular disease	This study estimated coronary vascular disease prevalence in 45+ individuals with chest pain or discomfort, using electronic health record data and expert panel consensus classification	Not mentioned	Not mentioned	Evaluate macrovascular and microvascular CAD in patients with chest pain	Decision-support tool development underway to aid in assessing coronary artery disease based on data [27,28]
Grober et al. [24], 2018	Cushing’s disease	The study evaluated pituitary MRI in Cushing’s disease patients using three techniques: DMRI, CMRI, and SGE, with anonymized annotations and independent reading by three clinicians	The study included 57 patients who had undergone surgery for Cushing’s disease; the patients consisted of 10 males and 47 females, with an age range of 13 to 69 years	Not mentioned	A blinded MRI of the pituitary gland in Cushing’s disease patients was performed	SGE showed better sensitivity for pituitary microadenoma identification in Cushing’s disease patients
Jalnapurkar et al. [21], 2017	Ischemic heart disease	A retrospective analysis of female patients with ischemia and no CAD who underwent stress cardiac magnetic resonance imaging from 2006 to 2007	The study analyzed 113 consecutive female patients who met the inclusion criteria	Not mentioned	Investigate stress CMRI in women with probable ischemia but no obstructive CAD	Subendocardial perfusion anomalies observed, indicating possible coronary microvascular dysfunction

**Table 2 diagnostics-13-02667-t002:** Machine learning application of CMRI for chest pain diagnosis.

Author/Year	Disease	Design of the Study	Number of Patients	Region	Aim	ML Method
Priya et al. [40], 2021	Pulmonary hypertension	A pilot radiomics study analyzed 72 CMRI images from 42 patients with pulmonary hypertension and 30 healthy controls, evaluating texture features’ diagnostic performance in predicting PH using various classifier models	The study analyzed CMRIs from a total of 72 patients, including 42 patients with PH and 30 healthy controls	Not mentioned	Classification	Using ensemble models (ridge, elastic net, LASSO, random forest, GBRM, AdaBoost) for non-invasive imaging-based recognition
Apfaltrer et al. [41], 2021	Cardioembolic stroke recurrence	A retrospective analysis of 151 patients with suspected cardioembolic stroke evaluated sensitivity, specificity, predictive value, and diagnostic accuracy using AUC	The study included a total of 151 patients with suspected cardioembolic stroke	Not mentioned	Prediction	Shapiro–Wilk test, chi-square test, the Fisher exact test
Iwata et al. [42], 2021	Myocardial ischemia				Classification	Decision tree
Wang et al. [43], 2021	Congenital heart defects	The study described in the article is a methodological study that proposes a myocardial segmentation algorithm based on adversarial learning; the authors conducted experiments using the HVSMR 2016 dataset to evaluate the performance of their proposed method	Not mentioned	Not mentioned	Segmentation	Generative adversarial networks (GAN), Gaussian mixture model, expectation maximization
Park et al. [35], 2020	Acute myocardial infarction	A retrospective study used machine learning algorithms to predict acute myocardial infarction in 4070 patients with chest pain, analyzing data from coronary angiography between 2004 and 2014	The study included a total of 4070 consecutive patients who had undergone CAG; the training set consisted of 3044 patients, while the test set included 1026 patients	Not mentioned	Classification	Using logistic regression, linear discriminant analysis, decision tree, K-nearest neighbor, support vector machine, and ensemble bagged tree
Sander et al. [31], 2020	Heart failure resulting from myocardial infarction, dilated cardiomyopathy, hypertrophic cardiomyopathy, and abnormalities in the right ventricle	The study utilizes automatic segmentation and uncertainty assessment in CMRI to detect local segmentation failures in cardiac structures, using publicly available scans from the MICCAI 2017 ACDC challenge	The article mentions that the complete set of scans of 100 patients was used for simulating manual correction of the detected regions; additionally, a random subset of scans from 50 patients was manually corrected	Not mentioned	Segmentation	Convolutional neural networks
Uthoff et al. [44], 2020	Pulmonary hypertension	The study developed a geodesically smoothed tensor (GST) feature-learning method for predicting pulmonary arterial hypertension using CMRI; it evaluated 150 individuals with confirmed PAH and 1-year mortality census, comparing GST method performance with RVESVi measurement	The study used CMRI scans from 150 individuals with confirmed PAH for evaluation; the article does not provide information on the comorbidities of the patients	Not mentioned	Classification	Geodesically smoothed tensor feature-learning method
Alis et al. [45], 2020	Patients diagnosed with hypertrophic cardiomyopathy may experience ventricular tachyarrhythmia	A retrospective study of 64 hypertrophic cardiomyopathy patients used machine learning classifiers to predict ventricular tachyarrhythmia (VT) presence using extracted features; the results were assessed using 24 h Holter monitoring and the synthetic minority over-sampling technique (SMOTE)	The study included a total of 64 patients with hypertrophic cardiomyopathy	Not mentioned	Classification	Support vector machines, naive Bayes, k-nearest neighbors, and random forest
Painchaud et al. [34], 2019	Delineation of the LV cavity, myocardium, and right ventricle from cardiac magnetic resonance images	The article presents a framework for cardiac image segmentation maps using CNN and cVAE, ensuring pre-defined criteria and inter-expert variability	Not mentioned	Not mentioned	Segmentation	Adversarial variational autoencoder, convolutional neural networks
Duan et al. [33], 2019	Pulmonary hypertension	The study utilizes multi-task deep learning and atlas propagation to develop a segmentation pipeline using a fully convolutional network architecture, incorporating 2.5D representation and refinement steps	The article states that the pipeline was validated on 1831 healthy subjects and 649 subjects with pulmonary hypertension; these numbers represent the patient cohorts used to evaluate the proposed method	Not mentioned	Segmentation	Shape-refined multi-task deep learning approach
Sparapani et al. [46], 2019	LV hypertrophy	The study used MESA cohort data to develop a new left ventricular hypertrophy criterion using Bayesian additive regression trees, comparing its diagnostic and prognostic performance with traditional ECG and imaging criteria	The analysis included 4714 participants from the MESA cohort	The study involved participants from MESA (Multi-Ethnic Study of Atherosclerosis), which is a multi-ethnic study conducted in the United States	Classification	Bayesian additive regression trees
Oktay et al. [32], 2017	UK Digital Heart Project dataset	The study introduces anatomically constrained neural networks (ACNN) as a training strategy, incorporating anatomical prior knowledge into convolutional neural networks; the approach is evaluated on multi-modal cardiac datasets and public benchmarks	Not mentioned	Not mentioned	Segmentation	Anatomically constrained neural networks, convolutional neural networks

## Data Availability

Not applicable.

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
