# Peer review of "Cardiac Magnetic Resonance Imaging (CMRI) Applications in Patients with Chest Pain in the Emergency Department: A Narrative Review"

_diagnostics, 2023, doi:10.3390/diagnostics13162667_

Round 1

Reviewer 1 Report (Previous Reviewer 2)

Thank you for your revision. I have no further comments.

Minor editing of English required.

Reviewer 2 Report (Previous Reviewer 1)

Dear Editor,

I really appreciate this new version of the paper. The manuscript is well written and updated.

This manuscript is a resubmission of an earlier submission. The following is a list of the peer review reports and author responses from that submission.

Round 1

Reviewer 1 Report

To:

Editorial Board

Diagnostics

Title: “Review of Cardiac Magnetic Resonance Imaging (CMRI) Applications in Patients with Chest Pain in the Emergency Department”

Dear Editor,

I read this paper and I think that:

-       Please revise the title: remove “review of” and include “a narrative review” at the end of the sentence.

-       Authors should revise the English of the paper due to typos. Please consider the help of a native English speaker.

-       The paper is really interesting but should deeply face the real world: how many cardiac MRI might they be performed in DEU? Cardiac MRI is time consuming and might delay interventions. Authors should dedicate an entire paragraph which should deal with the limitations of the application of cardiac MRI in daily clinical practice.

-       Furthermore, authors should provide data or speculation about the real impact of MRI rather than other imaging techniques in fast diagnosis and management of patients. Please update the work.

-       Figure 2, 3, 4, and 5 are redundant in the general background of the article. Authors can provide the same information into the paragraph thus avoiding figures. Please remove.

-       The contents of table 1 should be better resumed as they seemed to reproduce narrative text. This is misleading when reading the table.

-       Furthermore, there are no effective data on the clinical application of MRI in the acute setting. Table 1 reproduced works which are mostly in the chronic/subacute setting, i.e. in patients who presumably were admitted to the ward. I think that the messages are too generic and poorly reproducible.

Authors should revise the English of the paper due to typos. Please consider the help of a native English speaker.

Reviewer 2 Report

The authors evaluated the usefulness of cardiac magnetic resonance in patients with chest pain in the emergency department. The concluded that due to the imperfection of conventional methods in accurately diagnosing acute cardiac ischemia, patients are still being inappropriately discharged from the emergency department. 

I have the following concerns:

1. Computed tomography seems to have the priority over CMR in the majority of emergency department as a diagnostic tool for many emergency conditions.

2. What are the practical implications of the study?

3. What kind of studies were enrolled? Observational or RCT?

4. Region, design of the study, number of patients, their comorbidities should be included in table.

5. Bias in research should be described. 

Minor editing of language required.